# The Influence of Filters on EEG-ERP Testing: Analysis of Motor Cortex in Healthy Subjects

**DOI:** 10.3390/s21227711

**Published:** 2021-11-19

**Authors:** Ilona Karpiel, Zofia Kurasz, Rafał Kurasz, Klaudia Duch

**Affiliations:** 1Łukasiewicz Research Network—Institute of Medical Technology and Equipment, 41-800 Zabrze, Poland; 2Institute of Psychology, University of Silesia, 40-007 Katowice, Poland; zofia.kurasz@us.edu.pl; 3Independent Researcher, 40-007 Katowice, Poland; rizkurasz@gmail.com; 4Faculty of Science and Technology, Institute of Biomedical Engineering, Silesian Centre for Education and Interdisciplinary Research, University of Silesia in Katowice, 41-500 Chorzów, Poland; klaudia.duch@smcebi.edu.pl

**Keywords:** preprocessing data, somatosensory cortex, ERP, filters

## Abstract

The raw EEG signal is always contaminated with many different artifacts, such as muscle movements (electromyographic artifacts), eye blinking (electrooculographic artifacts) or power line disturbances. All artifacts must be removed for correct data interpretation. However, various noise reduction methods significantly influence the final shape of the EEG signal and thus its characteristic values, latency and amplitude. There are several types of filters to eliminate noise early in the processing of EEG data. However, there is no gold standard for their use. This article aims to verify and compare the influence of four various filters (FIR, IIR, FFT, NOTCH) on the latency and amplitude of the EEG signal. By presenting a comparison of selected filters, the authors intend to raise awareness among researchers as regards the effects of known filters on latency and amplitude in a selected area—the sensorimotor area.

## 1. Introduction

The EEG-ERP has become very popular, especially in the last 20 years, due to its many advantages: The examination is inexpensive, safe and easy to carry out. The modern EEG is considered to be an excellent alternative to magnetic resonance imaging or computer tomography because of its outstanding high temporal resolution [1]. The signal is widely applied in cognitive science and psychophysiological research, providing helpful clues about cognitive development and general mental functions [2]. The EEG is often used in neurological clinical trials to identify and treat various diseases, such as epilepsy [3], sleep disorders [4,5] and Alzheimer’s disease [6]. The EEG can also provide plenty of useful information about various mental dysfunctions or mental illnesses [7,8]. This is why it is gaining ground in psychological and psychiatric research and diagnosis in areas such as schizophrenia [9,10], Internet addiction [11], bipolar disorder [12], depression [13], binge eating disorder [14], different personality disorders [15,16] and many others.

Owing to the growing popularity of the EEG over the past two decades, researchers have definitely been using more instruments. In turn, the development of new tools and software has made it possible to conduct more complex and methodologically advanced research. This is closely related to the demand for knowledge about data processing. There is growing pressure on developing more advanced statistical and signal processing methods. In addition, scientific practice is currently undergoing tremendous changes to improve the transparency of data collection, documentation and analysis, research reproducibility and manuscript review [17]. Thus, a unified approach must be applied to data recording, artifact control and signal processing, and the procedures must be reported transparently.

However, a review of the literature shows that this is not an evident approach. As summarized by Robbins et al. [18], to date, studies have differed significantly already at the stage of data pre-cleaning methods, and EEG preprocessing approaches have not been standardized. Basically, the preprocessing phase is essential because a raw EEG signal is always contaminated with many different artifacts. They are related to the subject, e.g., as muscle movements, eye blinking, cardiac activity or breathing, or to other external and environmental factors, such as powerline disturbances [19]. Different methods are used to remove each type of artifact.

The methods are available as built-in options in various toolboxes or as dedicated overlays which support the preprocessing phase. EEG signals recorded by a single electrode are a superposition of signals from multiple sources (e.g., assigned to other electrodes); therefore, so-called spatial filters have come into use. There are also methods based on the principle of source signal estimation. An example of such a method is Blind Signal Separation (BSS) and, in particular, Independent Component Analysis (ICA) (Delorme 2004). One of the more popular environments used for preprocessing is MATLAB software together with open packages, i.e., EEGLAB, ERPLAB [20], SPM or Fieldtrip, or commercial ones, i.e., Curry or BESA. The aforementioned filters have been widely described in scientific papers both in technical terms, FIR [21,22], IIR [23,24], FFT [25,26], NOTCH, and in clinical applications or in typical research, FIR [27], NOTCH [28], FFT [29].

There is a recommendation for a package like EEGLAB that continuous EEG data should be filtered before being epoched or before the artifacts are removed. The process of filtering continuous data minimizes the need to filter artefacts at epoch boundaries (the use of FIR [30] is “recommended”). In the absence of the MATLAB Signal Processing toolbox (the EEGLAB legacy filter), a simple filtering method by means of the inverse Fourier transform can be applied. However, there are more filtering options. There is no ideal filter for the EEG data, either. For example, the effect of high-pass filters on ERP data is currently under discussion. Consequently, different solutions can be found in the literature: Both without filtering (e.g., VanRullen, [31]) or with very careful filtering [32]. There is an ongoing discussion in the literature [30,32,33,34] on how to avoid suboptimal practices resulting in regularly distorted or even false results.

External noises are removed at the first stage of data processing with the use of filters designed for this purpose. However, it is by no means easy to choose the right filters, as there is no adequate research into comparing the filtration effect by means of different filters. While several guidelines for best practices in signal preprocessing have been released [17], the procedures are relatively broad and give researchers considerable leeway to choose the appropriate filters or to create processing pipelines (in some parts, manual) consistent with best practices [18].

The absence of specific standards at the stage of preliminary signal purification and preparation is a cause for concern. To maintain a specific frequency, the signal needs to be cleaned with filters that should meet strict requirements. Any distortion of the original EEG signal can have fatal consequences, both for research purposes and in clinical use, at the stage of diagnosis and medication. Unfortunately, for the time being, little is known about the extent to which the use of different filters ultimately makes a difference in the obtained data. This refers in particular to latency and amplitude, which are important parameters for neuroscience and clinical diagnostics. There is not enough literature to compare various techniques used for artifact rejection. A standard methodology for perfect artifact removal has not been defined [35]. What is more, even those studies that strictly adhere to guidelines exhibit significant differences in the application of the recommended methods. This is a serious issue if the results between different studies are compared [18]. Another issue is that few researchers are familiar with certain artifact detection and removal methods [36], even though they are universally available [37].

The paper attempts to determine the influence of the filters on the EEG-ERP study, in particular the influence of the selected filters on the amplitude and latency in the sensorimotor area.

## 2. Materials and Methods

The data for the analyses were reused and the experiment itself was described in an earlier paper by Karpiel and Drzazga [38].

### 2.1. Participants

Seven healthy participants (4 males and 3 females) aged 20 to 35 (mean = 24.14, SD = ±4.99) without any neuromuscular or cerebral diseases took part in the study as volunteers. They were university students or alumni. All participants were right-handed, had normal color perception and normal visual acuity. All of the participants were healthy and had no neurological medical history. Information about their health and lifestyle was collected via a survey. The study was approved by the institutional ethics committee of the University of Silesia.

### 2.2. Experimental Design

Raw EEG signal data were taken from an experiment conducted in the laboratory. Participants were shown three Arabic numerals: 2, 3 and 4. Their task was to look at the center of the screen and respond to these numerals by pressing the appropriate keys on the keyboard with the index finger, middle finger and ring finger, respectively. A total of 450 right-hand and 300 left-hand trials were made. The stimulus interval was set at 800 ms. Each complete trial (stimulus + response + pause) lasted 2 s. Therefore, it took 15 min to complete the task for the right hand and 10 min for the left hand. Scenarios were created with Eevoke software (ANT Neuro running on PC desktop—MS Windows 7 OS). The experiment was carried out with a commercial ANT Neuro amplifier (AMP-TRF40AB model) in DC with 20,000 amplification gain and 256 Hz sampling rate, as well as ASA v.4.8 software. The stimuli included the Arabic numerals: 2, 3 and 4. The raw EEG signal and reaction times were recorded during the process for later analyses. The participants were seated in a comfortable chair, in front of a computer screen, at a distance of 1 m.

### 2.3. Experimental Procedure

Digits 2, 3 or 4 were shown in the center of the screen until a key was pressed, or until the digit automatically disappeared after 1200 ms. The subjects were required to respond by pressing the key with their index finger when “2” appeared, the middle finger when “3” appeared and the ring finger when “4” appeared. A new stimulus arrived 800 ms after the previous one. The order of all attempts was randomized. There were 150 attempts for each of these three arrangements for the right hand, and 100 attempts for the left hand. The order of these attempts was randomized.

### 2.4. Electroencephalogram Acquisition and ERP Recording

The EEG signal was recorded according to the standard 10–20 system, using an ANT Neuro amplifier and the ASA software, v.4.8. However, only eight electrodes were selected for the filtering process: Two frontocortical (F3, F4), two centrocephalic (C3, C4), two parietal (P3, P4) and two occipital (O1, O2). These channels were chosen because the visual perception and motor response had to be analyzed.

The raw EEG signal from 450 measurements for the right hand and 300 measurements for the left hand was refined four times; a different filter was used each time. Subsequently, the ICA analysis was intended to remove the artifacts, divided into 450 and 300 epochs, respectively, and averaged. The maximum amplitude and latency were determined from the averaged signals. At the final stage, the amplitude and latency values of the raw signal and the signals purified by filters were compared. The raw signal was filtered with the use of four different filters (FIR, IIR, FFT (0, 5–40 Hz) and NOTCH (lower passband edge 46.5 Hz, upper passband edge 127.5 Hz)) and processed. Two parameters are used to compare the effect of the filter on the EEG signal: Latency and amplitude. All analyses were performed in Python using the MNE toolkit on a PC. All analyses were performed using PC Dell Latitude E7270 with CPU Intel Core i5-6300U, Intel HD Graphics 520, 8 GB RAM and Windows 10 64 operating system. The data will be used to propose a model which uses machine learning [39,40].

### 2.5. Statistics

A statistical analysis was performed with the Statistica 13.1 software. For all measures, descriptive statistics were calculated. The Shapiro–Wilk test was used to check the normality of distributions of the studied variables. The homogeneity of variances in the analyzed groups was verified by Levene’s test. If the data failed to fulfill the assumptions required for a parametric test, a nonparametric Friedman’s ANOVA test was applied. The level of significance was set at *p* < 0.05.

## 3. Results

First, P100 and P300 refractions were located to verify the brain region in which the highest values were obtained. To present the results, we focused on selected electrodes/areas and the maximum and minimum values of both latency and amplitude. Several comparisons were made to show the differences between the selected approaches.

Figure 1 shows the averaged ERP curves of all three fingers from the O1, O2, Oz leads for seven tests. In the early visual phase (P75-120), the mean amplitude of O1 was the strongest. In the pre-execution phase (N175-260), the mean amplitude of O2 was the strongest. In the execution phase (P310-420), the strongest mean amplitude was for Oz (see Figure 2). The plot shows the envelopes (i.e., the min and max values, over all channels, at each time point), where the displayed topomaps of the average field in 50 ms time window centered at 129 ms, 172 ms and 375 ms. On the basis of the collected waveforms and topomaps, the highest values of amplitudes and latencies were selected. The figure shows that in the range between 0.1 and 0.2, the highest amplitude values of ~9 µV were obtained in the visual and motor cortex areas.

Figure 2 depicts the averaged curves of all three fingers from O1, O2 and Oz leads for seven tests. In the early visual phase, the highest mean amplitude value can be observed for the O1 lead with a slight advantage over the Oz and O2 leads.

Based on statistical analyses, there were no significant age-related differences for either latency or amplitude. By combining these topographical and temporal amplitude analyses, we confirmed the presence of differential responses to touch.

### 3.1. Latency and Amplitude—Influence of the Filters

EEG–ERP recordings were analyzed in order to identify the evoked potentials. On the basis of signal recordings obtained from seven subjects, tables were prepared with the values of amplitudes and latencies for the electrodes. Table 1 contains information on the analysis. Details concerning the group are included in Table 2. In the table for each subject there is a division into the first and second paradigm (I and II), type of filter and maximum and minimum values of latencies and amplitudes for selected electrodes.

It should be emphasized that among a considerable amount of data, only maximum and minimum values were chosen for the selected number of electrodes. In the first row, the highest amplitude value (for further consideration) is presented, which is associated with electrode C4 and exhibits the value of 250 ms. The latency was handled in the same way. Electrodes O1 and O2 were omitted in the presentation to narrow down the selected area to the sensorimotor area in the best possible way.

Our analyses showed that the lowest latency obtained was 250 ms (C3, C4, P3, P4) and the highest latency value was 664 ms (F3). The highest latency value was observed in the signal row for electrode F3, for the test performed with the left hand. The lowest band pass latency obtained was 250 ms (C3, C4, P3, P4) and the highest was 664 ms (F3). The analysis for “notch” also showed the lowest latency value of 250 ms (C3, C4, P3, P4). In contrast, the highest value was 664 ms (F3) (Table 1 and Table 2).

### 3.2. Differences between Filters

The analysis was divided into a first paradigm analysis (for the right hand) and a second paradigm analysis (for the left hand), respectively.

First paradigm. If analyzed sequentially, for one patient, the maximum value of latency changed/increased, the lowest was for raw, and the same for pass band and notch (648 ms). In patient 2, no difference was seen in the filters used; the latency value did not change. For the next one (3), the lowest latency value was 296 ms (C3) for raw, higher for pass band and even higher for notch, while the maximum latency value, interestingly, was the highest for raw (597 ms) and much lower for the other filters (382 ms). In case 4, the minimum latency value was the same for all. The maximum latency value for raw was obtained for C3 and was slightly higher for the other filters (664 ms) and in F3. In the analysis of patient 5, no difference was noticed in the filters used; the latency value did not change. In the analysis of patient 6, an increasing trend was observed, both for the minimum and maximum latency value. However, in the analysis of patient 7, the minimum value of latency (raw signal) for three electrodes (F3, C3, C4) was obtained, and it was exactly the same value, slightly bigger than for the applied filters. The maximum latency value for raw and filters was the same.

Second paradigm. A detailed analysis for each individual is as follows: For patient 1 the minimum and maximum latency value for all analyses is the same. For patient 2, the minimum latency value is 312 ms (C3), while for the filters it is the same at 328 ms. Taking into account the maximum value of latency for raw, it is much higher at 542 ms (P4), and for the analysis using selected filters it is only 398 ms (F4). For patient 3, the minimum latency value for raw and filter is the same, while the maximum is the same for raw and notch at 648 ms (C4), and lower for band pass at 519 ms (F4). An analysis of patient 4 indicates that the minimum latency value for raw and filters is the same at 250 ms (C3, C4, P3, P4), while the maximum is the same for band pass and notch. The raw signal analysis is marginally higher. It can be seen in the case of patient 5 that the minimum latency value for raw and filter is the same, while the maximum latency value for pass band is slightly higher than the other values. For patient 6, the minimum and maximum latency value for raw signals is the highest. Patient 7 exhibited no difference in the values between the minimum latency values for the analyses, while the maximum was the highest for notch.

Summary: Filters can affect the final latency time. They can change the latency values in comparison with the raw signal. In five out of seven patients, greater maximum latency values were observed for the first paradigm used compared to the second paradigm used for the left hand (filters). For raw signals: For four out of seven cases, higher maximum latency values were observed for the first paradigm used compared to the second paradigm used for the left hand.

### 3.3. Amplitude and Latency for Fingers 2, 3 and 4

Next, a more detailed analysis was performed. For each of the eight electrodes, data concerning the amplitude and latency were gathered. The values concerned all fingers, 2, 3 and 4, focusing on the raw signal, FIR, IIR, FFT and NOTCH. The analysis was performed both for the right and left hand. Table 3 exhibits data for the right hand for one patient.

As shown in Table 3, for all electrodes, different values were obtained for the unfiltered signal compared to the other filters. Each record under analysis produced a fairly large numerical database. It can be inferred from Table 3 that for the IIR filter the amplitude for electrode O1 was 4.02 and for the rest 4.66. A slightly higher amplitude, considering the selected electrodes, is noticeable for the NOTCH filter. On the other hand, slightly lower latency (for all electrodes) has been detected for the NOTCH filter in comparison to the other presented filters. FIR, IIR and FFT in this particular case have the same values of latency and amplitude. Interestingly, no regularities were observed to predict how amplitude or latency would change.

Figure 3, Figure 4, Figure 5, Figure 6 and Figure 7 exhibit comparisons of amplitude and latency values for each filter (divided into the left and right hand). Figure 4 and Figure 5 give a full picture of how latency and amplitude change. The presentation of the graphs includes data for both paradigms, i.e., for the right and left hand. Both Figure 4 and Figure 5 are box and whisker plots, showing the results for the applied FIR filter and its effect on the data averaged for all fingers, and for fingers 3 and 4, respectively. When analyzing the amplitude for the right hand movement (left panel), we can see the biggest scatter for electrode P4 and the smallest for O1. For the left hand, the largest scatter was obtained for electrode C4 and the smallest for electrodes F4 and P4. A graph was drawn for electrodes F3 and F4 showing the difference in the obtained latency values. The smallest differences were found for the right and left hands for the FFT, FIR and NOTCH filters (Figure 3).

The latency analysis in Figure 5 shows the smallest scatter for electrodes O1 and O2 and a much larger scatter for the other electrodes. There is a noticeable difference between the latency values for finger 4. If we compare the two paradigms, a smaller scatter is observed for electrodes P3 and P4 for the left hand compared to the right hand.

Comparisons of all selected filters were made. The graphs can be read twofold. Latency and amplitude values can be compared for each of the fingers, for the right and left hands for the selected electrodes. In addition, the values for the selected filters are presented. Consequently, (comparing the results for individual fingers) for the right hand, the largest scatter was noticed for finger 4. The difference between the filters used was negligible. For the left hand, the largest scatter was for finger 3. Interestingly, in this analysis there were differences in the filters used. For example, for FIR there was relatively little scatter for finger 2 compared to the IIR used.

For the FIR filter, the amplitudes and latencies can be seen from the individual electrodes. In the case of O1 and O2 electrodes, the lowest latency values can be seen (Figure 6 and Figure 7).

The differences between the right and left hand are noticeable; for example, the FIR filter, both for finger 3 and finger 4. For finger 3 in the left hand, the scatter is significantly larger compared to the right hand; the same goes for finger 4.

For the right hand, the largest scatter can be seen for finger 4, while for the left hand, the largest scatter is discernible for finger 2. When considering the right hand, there is not much difference between the filters used. For the left hand there are noticeable differences between the filters used. In particular, a larger scatter can be observed for fingers 2 and 3 compared to the other filters. If we compare the latencies for the right hand (movement of all fingers) and the chosen electrodes, the selected significance level is less than *p* < 0.05 and amounts to *p* = 0.00009 (for notch). For the analysis performed on the second finger *p* = 0.00006, third *p* = 0.00004 and fourth *p* = 0.00004. However, taking into account the amplitude (movement of all fingers, and only electrode C3), *p* = 0.03.

## 4. Discussion

There are more and more tools and possibilities for data analysis. The methods used can significantly affect the results and thus the subsequent diagnosis. Therefore, more papers should be published on different comparisons of software, methods and algorithms in use. This would make it much easier for young researchers at the beginning of their medical career to learn how to analyze and interpret the obtained results. There are over 150,000 publications on ERPs alone in the pub med database and over 17,000 on sensorimotor ERPs. This vast amount of information shows that it is often difficult to compare the results because they may depend on the methods used. The EEG itself is a relatively broad topic, yet new papers are published, related to the study of motor or speech areas, which are also key to understanding the motor planning of speech [41] or more broadly discussed sensory attenuation and the involved phenomena [42,43]. A few years ago, papers were also published with a focus on presenting bioelectrical results in the area of sensorimotor cortex not only for diseased subjects [44], but also for healthy/control subjects [45].

It seems to the authors that far too few papers have been published with a focus on comparing several methods of analysis, or based on several different parameters. This seems to be a niche among the original papers. The impact of filters on data interpretation is not always fully appreciated. This paper reviews this issue and explains how a particular filter affects the ERP data if at all. The theoretical part related to the construction and description of filters, and to the problems that can be expected when using them, to the method of choosing the right filter and how to avoid filtering by using alternative tools has been elaborated on in the paper by Alain de Cheveigne and Israel Nelken [46].

Filters are ubiquitous in brain data measurement and analysis. All data scientists involved in the analysis of both ERP and other signals deal with preprocessing and in particular with the choice of a filter for a specific analysis. It is important to improve the SNR, but it must also be remembered that filters affect the target signal. The authors have tried to point this out in this work.

Obvious as this should be, most papers provide incomplete information about the filters used and the “methodology” in general due to journal restrictions. Another issue is the “arbitrariness” of the procedure and analysis, since the preprocessing alone can be done arbitrarily, with different parameters.

The literature provides examples related to the use of filters, where it is indicated that their role is to suppress noise and enhance target activity. Yet, they can have detrimental effects that the researcher should be aware of.

Considering the overall number of emerging publications related to image preprocessing (or based on image preprocessing) or even signals, relatively few papers present a comparison of results. By this we mean a comparison involving exactly the same data, but including a comparison of different types of analysis, or a change of one or more parameters. The results are more often presented using one of the adopted models of the preprocessing analysis.

The question remains, can changes in amplitude or latency be crucial to the performance of EEG-based systems? Rhythms are divided primarily by their frequency range, but also by their amplitude, shape, duration and the location of the brain activity that accompanied their generation. Therefore, any change in amplitude or latency due to preprocessing or later postprocessing can have a significant impact on the medical expert’s assessment.

The authors focus on presenting the results concerning the sensorimotor area, which is of great importance in the diagnosis related to neurodegenerative diseases, such as patients with multiple sclerosis [47,48], Parkinson’s or Alzheimer’s disease [49]. The value of both amplitude and latency is of great significance as attention is paid to each value in the context of a given disease. For example, the usefulness of evoked potentials in the diagnosis and monitoring of MS is widely described in the literature [47,50]. Considering a prolongation of P100 latency or a decrease in amplitude below 5 μV, it significantly changes the diagnostic process. Moreover, visual evoked potentials are useful neurophysiological methods in the diagnosis of subclinical demyelinating foci in the central nervous system (CNS) and the presence of changes in somatosensory evoked potentials indicates the need for a thorough diagnosis of spinal localization in MS.

Some of the waveforms, particularly P50, N100, MMN, P300 and N400, are proposed as biomarkers of schizophrenia [51]. A distinction is made between state markers (episodic, relating to symptoms) and traits (persisting independently of the clinical status) [52]. Based on the results of the study, it is proposed that visual P300 potentials may be a marker of the clinical condition of schizophrenia (research results indicate i.a. that visual P300 potentials are an indicator of the severity of clinical symptoms) [53,54,55], while auditory P300 potentials—a marker of trait or susceptibility to schizophrenia [56].

Somatosensory evoked potentials permit objectification and determination of the location of sensory disturbances. They can complement electronurography and electromyography (EMG, electromyography) in the diagnosis of peripheral nervous system lesions (carpal tunnel syndromes, brachial plexus injuries, upper chest syndrome). They are also used in the diagnosis of spinal cord injuries (trauma, myelopathy, demyelinating disorders). Moreover, they play an important role in intraoperative monitoring of the integrity of sensory pathways (spine and spinal cord procedures) [57]. Similar to BAEP, somatosensory evoked potentials are used to assess brainstem function in unconscious patients [58].

## 5. Conclusions

Filters form an important part of any research methodology. They are designed to amplify the signal and suppress noise but, apart from their undeniable advantages, they can cause harmful effects. The user should be aware of the impact of filtering, particularly in the field of neuroscience. Based on seven research examples, it can be concluded that significant differences in both amplitude and latency values can occur due to the filter used and the preprocessing performed. It seems that there is a need to “standardize” preprocessing and filtering for individual disease entities in order to avoid erroneously prepared data, which, in turn, may result in a wrong medical diagnosis. The reader, researcher or expert should be aware of the potential effects, therefore more reports should be provided for different areas and selected potentials.

In conclusion, it seems that each analysis should be performed at least twice to verify the results and to minimize the likelihood of misdiagnosis. To the best of our knowledge, the preprocessing of applied filters and other values is not structured yet, and there are simply some recommendations related to, e.g., the software used. Therefore, the literature exhibits quite a lot of freedom in the way that the selected methods are used. Each method has its supporters and opponents. Even an expert is unable to make a proper diagnosis without reliably prepared data. Hence, the influence of the filters on the amplitude and latency calls for a detailed analysis.

## Figures and Tables

**Figure 1 sensors-21-07711-f001:**
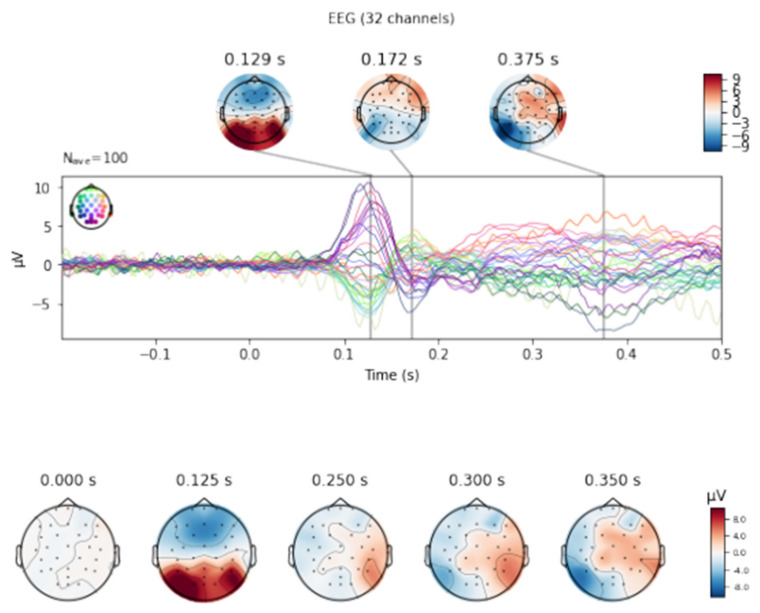
Examples of waveforms. Butterfly plot of somatosensory evoked potentials for all EEG channels (colored lines, 32 channels). The averaged EEG channels are marked in the drawing of the scalp next to the y-axis. Below: Topographical distribution of each component on the scalp. Red and blue indicate the maximum and minimum EEG amplitude at each time point, respectively.

**Figure 2 sensors-21-07711-f002:**
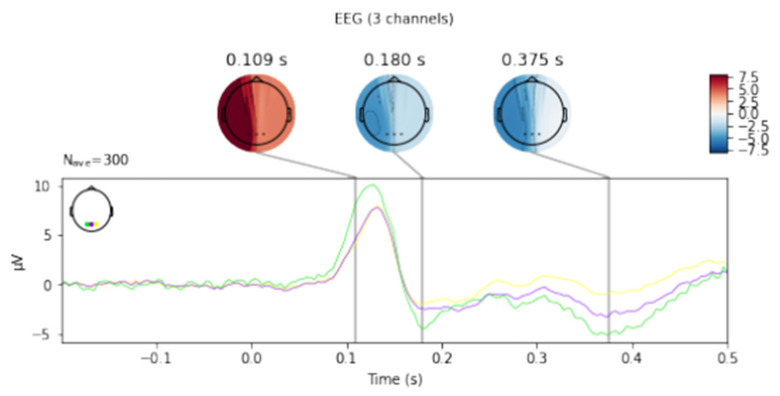
The averaged signal for electrodes O1 (green line), Oz (purple line) and O2 (yellow line).

**Figure 3 sensors-21-07711-f003:**
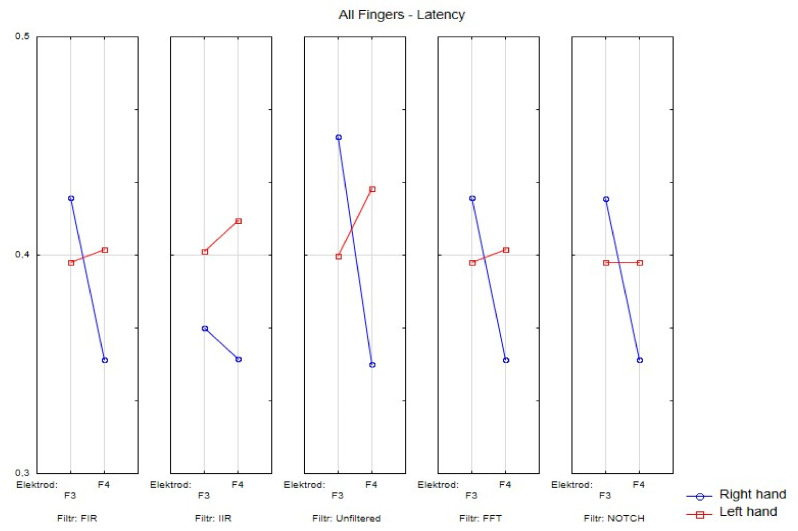
Presentation of the average latency values for the F3 and F4 electrodes for the right and left hands for selected filters.

**Figure 4 sensors-21-07711-f004:**
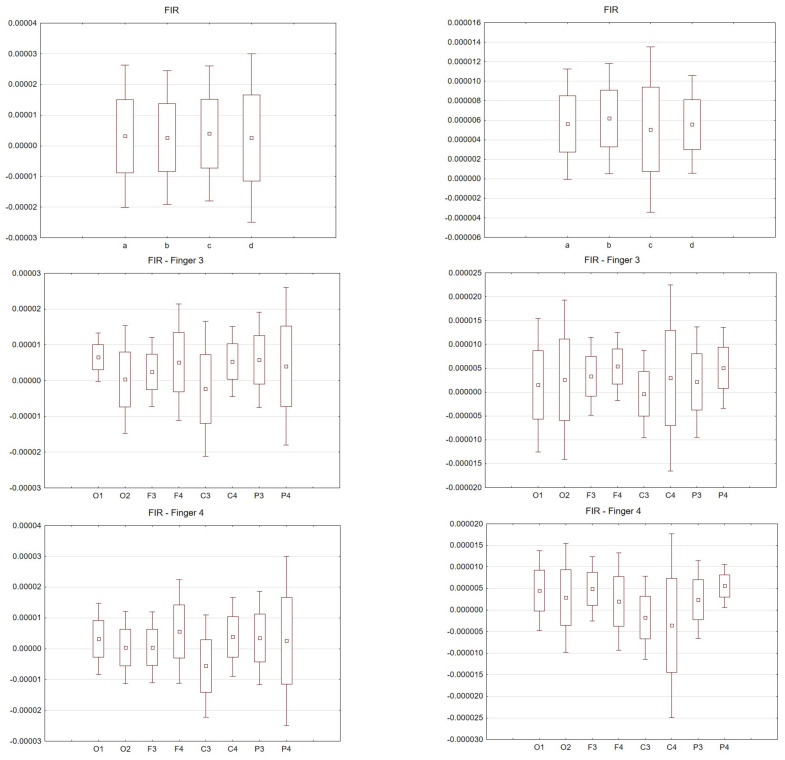
Wave amplitude values (V) FIR filter: Top drawing for all and individual fingers: a—all fingers, b—“2” fingers, c—“3” fingers, d—“4” fingers and electrodes P4. Middle drawing for finger 3 and all electrodes. Bottom drawing for finger 4 and all electrodes. Right hand (left in the photo) and left hand (right panel).

**Figure 5 sensors-21-07711-f005:**
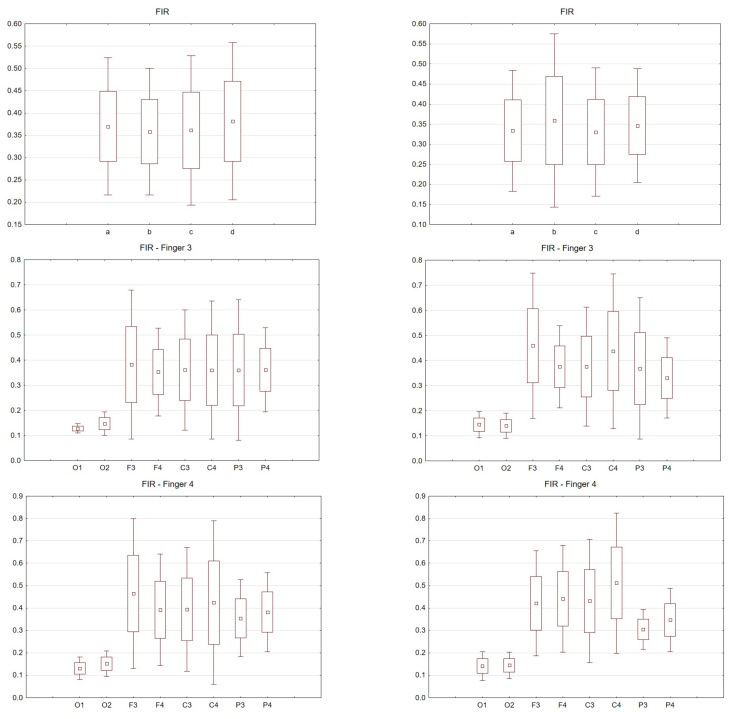
Latency values (s) FIR filter: Top drawing for all and individual fingers: a—all fingers, b—“2” fingers, c—“3” fingers, d—“4” fingers and electrodes P4. Middle drawing for finger 3 and all electrodes. Bottom drawing for finger 4 and all electrodes. Right hand (left in the photo) and left hand (right panel).

**Figure 6 sensors-21-07711-f006:**
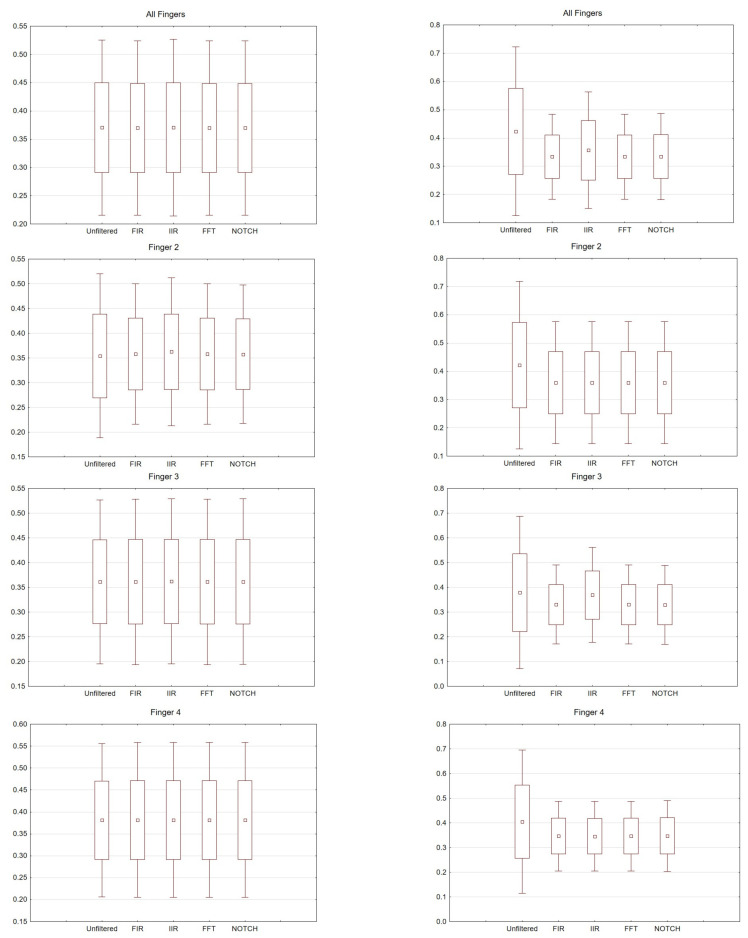
Box and whisker plot depicting latency values for averaged values, fingers 2, 3 and 4. From the left: Right hand and left hand.

**Figure 7 sensors-21-07711-f007:**
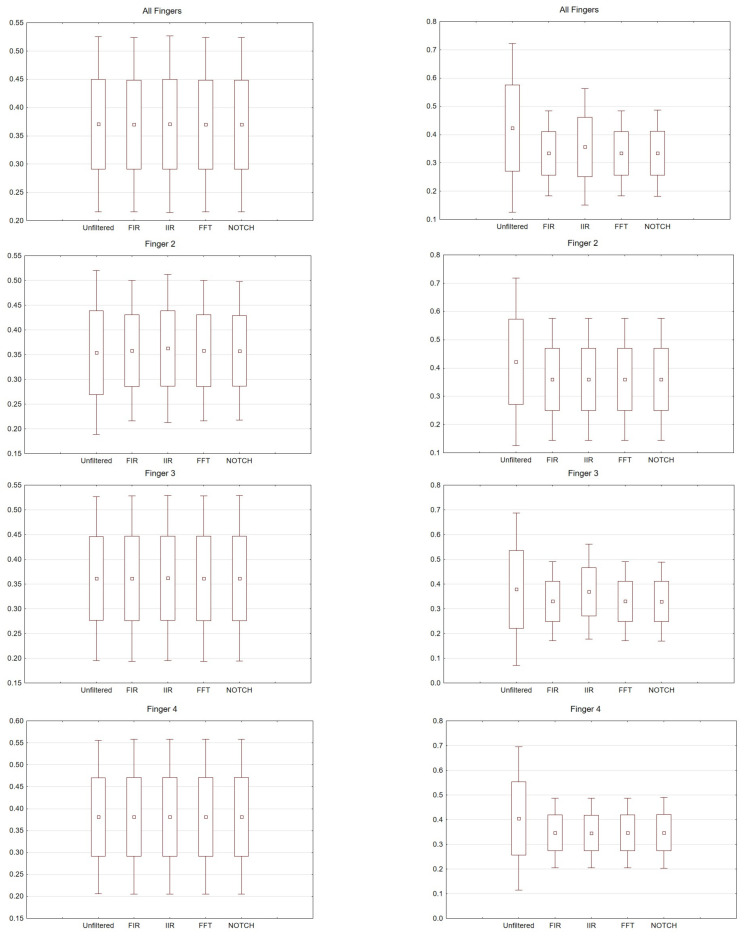
Box and whisker plot depicting amplitude values for averaged values, fingers 2, 3 and 4. From the left: Right hand and left hand.

**Table 1 sensors-21-07711-t001:** Latency values for seven subjects, for selected electrodes (C3, C4, P3, P4, F3, F4).

No	Time	Filter	Latency max (ms)	Electrode	Latency min (ms)	Electrode	Amplitude max (µV)	Electrode	Amplitude min (µV)	Electrode
1	I /before 12 a.m.	raw	593	C4	250	C3	4.16	P4	−4.48	P3
		pass band	648	C4	250	C3	4.10	P4	−4.41	P3
		notch	648	C4	250	C3	4.09	P4	−4.41	P3
	II /after 12 a.m.	raw	371	P4	250	C3	4.01	C4	−2.18	P3
		pass band	371	P4	250	C3	3.86	C4	−2.18	P3
		notch	371	P4	250	C3	3.89	C4	−2.16	P3
2	I /after 12 a.m.	raw	445	C4	371	P3	21.4	F4	−20.5	C3
		pass band	441	C4	371	P3	21.1	F4	−20.2	C3
		notch	441	C4	371	P3	21.1	F4	−20.2	C3
	II /after 12 a.m.	raw	542	P4	312	C3	10.7	F4	−9.29	C4
		pass band	398	F4	328	C3	10.6	F4	−9.27	C4
		notch	398	F4	328	C3	10.6	F4	−9.26	C4
3	I /before 12 a.m.	raw	597	F3	296	C3	9.66	C4	−11.1	P4
		pass band	382	F3	312	C3	9.32	C4	−10.7	P4
		notch	382	F3	316	C3	9.36	C4	−10.7	P4
	II /before 12 a.m.	raw	648	C4	328	C3	5.22	F3	−4.65	C3
		pass band	519	F4	328	C3	5.10	F3	−4.14	C3
		notch	648	C4	328	C3	5.25	F3	−4.20	C3
4	I /after 12 a.m.	raw	0.66015625	C3	250	F4	15.4	P4	−8.43	C3
		pass band	664	F3	250	F4	15.4	P4	−8.35	C3
		notch	664	F3	250	F4	15.4	P4	−8.37	C3
	II /after 12 a.m.	raw	371	F3	250	C3	24.2	C4	5.68	F3
		pass band	367	F3	250	C3	24.1	C4	5.59	F3
		notch	367	F3	250	C3	24.2	C4	5.63	F3
5	I /after 12 a.m.	raw	644	F3	296	C4	7.42	P3	−12.0	P4
		pass band	644	F3	296	C4	7.38	P3	−11.9	P4
		notch	644	F3	296	C4	7.37	P3	−12.0	P4
	II /after 12 a.m.	raw	546	C4	304	P4	6.99	P4	−4.33	C4
		pass band	550	C4	304	P4	6.88	P4	−4.30	C4
		notch	546	C4	304	P4	6.91	P4	−4.27	C4
6	I /after 12 a.m.	raw	406	C3	265	F3	2.14	F3	−2.05	C3
		pass band	433	C3	257	C4	1.97	F3	−1.58	C3
		notch	433	C3	257	C4	1.98	F3	−1.58	C3
	II /after 12 a.m.	raw	664	F3	335	P3	5.52	P4	−2.22	C3
		pass band	660	F3	250	P4	5.20	P4	−2.07	C3
		notch	660	F3	250	P4	5.29	P4	−2.11	C3
7	I /after 12 a.m.	raw	507	P3	285	F3	8.17	C4	0.363	P3
		pass band	507	P3	281	F3	8.10	C4	0.371	P3
		notch	507	P3	281	F3	8.16	C4	0.360	P3
	II /after 12 a.m.	raw	355	F3	289	P3	6.03	P4	−2.10	P3
		pass band	359	F3	289	P3	6.04	P4	−2.07	P3
		notch	363	F3	289	C3	6.06	P4	−2.08	P3

**Table 2 sensors-21-07711-t002:** Specification of the group selected for further analysis which takes into account age, gender and whether the person is right- or left-handed.

No	Age	Sex	Hand
1	20	M	R
2	22	F	R
3	35	M	R
4	27	F	R
5	20	F	L/R
6	21	F	R
7	24	M	R

**Table 3 sensors-21-07711-t003:** Numerical values (latency and amplitude) for finger 2, for the right hand, for one person.

Filter	Electrode	Latency (ms)	Amplitude (µV)
RAW	O1	148	5.3
RAW	O2	156	2.38
RAW	F3	265	3.78
RAW	F4	347	−1.23
RAW	C3	535	−2.75
RAW	C4	265	−1.95
RAW	P3	253	1.51
RAW	P4	351	−1.95
FIR	O1	152	4.66
FIR	O2	152	2.22
FIR	F3	269	2.98
FIR	F4	359	−1.34
FIR	C3	433	−2.09
FIR	C4	250	−1.64
FIR	P3	320	1.44
FIR	P4	324	−1.69
IIR	O1	152	4.02
IIR	O2	152	2
IIR	F3	269	2.57
IIR	F4	359	−1.09
IIR	C3	429	−1.72
IIR	C4	250	−1.45
IIR	P3	320	1.28
IIR	P4	324	−1.55
FFT	O1	152	4.66
FFT	O2	152	2.22
FFT	F3	269	2.98
FFT	F4	359	−1.34
FFT	C3	433	−2.09
FFT	C4	250	−1.64
FFT	P3	320	1.44
FFT	P4	324	−1.69
NOTCH	O1	148	4.6
NOTCH	O2	156	2.18
NOTCH	F3	265	3.1
NOTCH	F4	359	−1.2
NOTCH	C3	464	−2.18
NOTCH	C4	250	−1.57
NOTCH	P3	320	1.45
NOTCH	P4	324	−1.7

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
