# Peer review of "The Influence of Filters on EEG-ERP Testing: Analysis of Motor Cortex in Healthy Subjects"

_sensors, 2021, doi:10.3390/s21227711_

Round 1
Reviewer 1 Report
In general, this paper does not have any significant drawbacks, it is editied cleanly, the research problem and the methodology and results are fine.
Initially I thought that this is just another paper which deals with artifacts removal, filtration, etc. and there are some many papers in this area that it is difficult to do something new. In this light I find it (to a level) interesting that the Authors managed to show the forementioned problems from the point of view of amplitude / latency analysis.
My only criticism is regarding Fig 4 and Fig 5 - I am not sure but maybe the Authors could find a bit more obvious way of showing these results. For now there is not too much difference between those graphs and at the first glimpse nothing is obvious (additionally, the axes description is very small font and not very informative). So I would expect to make some improvements in this part.
Use of English is fine, just one proof read would do.
Author Response
Responses to Reviewer 1 Comments:
Thank you for all the comments.
I used the "Track Changes". Changes are visible throughout the document.
Changes: I edited the entire document for comments from all reviewers.
The paper was proofread by a certified linguist.
I changed figures 4 and 5. I hope that now the presentation of the results is better.
Reviewer 2 Report
Introduction section must state for which practical or experimental cases the changes of Amplitude in the EEG signals could be affect the experimental results. That is, I consider that filtering is an essential technique to minimize or eliminate undesired noise or signal perturbations, but consequently, filters could attenuate the original signal. However, we use filters because this attenuation in the signal amplitude (or the increment of latency) can be handled and do not tightly affect the final results. So, my point is, when this changes in amplitude or latency can be crucial for EEG-based systems performance?
Line 53 sets that different methods are used but how often the selected filters (FIR, IIR, FFT or NOTCH) are really used? Select the common approaches that use these filters, to highlight contributions of the proposed work. In Results section, Figure 1 is referred many times to confirm some assumptions about strongest mean amplitude, however, this figure does not explain where is Oz, from O1 or O2, even O1 is also referred as 01. For clarity, I consider that these legends must be included in the figure. Furthermore, the approximation of the classification accuracy in line 150, refers once again Figure 1, but it is unclear how to interpret “the classification accuracy” on the figure. I don’t understand the objetive of including lines 154, 166-167. Figure 2 and 3 are missing, although these figures are not referred on the text. Typing mistakes must be corrected along the document.
Experimental design and procedure are adequately, however figures should be clearly explained, the PC system used to measure latency must be included. Also, authors should highlight practical applications for which filtered signals could throw lower results than unfiltered results. That is where Amplitude of EEG signals measure important cognitive activities in the brain.
Author Response
Responses to Reviewer 2 Comments:
Thank you for all the comments.
I used the "Track Changes". Changes are visible throughout the document.
Changes: I edited the entire document for comments from all reviewers.
The paper was proofread by a certified linguist.
It may seem to readers that this is just another paper which deals with artifacts removal, filtration, etc. There are some many papers in this area that it is difficult to do something new. However the Authors to show the forementioned problems from the point of view of amplitude / latency analysis.
The impact of filters/software/analysis is not often presented and specifically considering ERP and motor cortex area. I wanted more researchers to be able to see the results and refer to their analyses or future study. I personally miss this type of publication, hence the idea to present the analysis in this way.
Changes:
Frequency of filter use and application : I modified the intruduction and discussion. I added the literature. I changed the presentation of the results. I added figure 2 to delineate the data. Legend is- left and right (colors, scale, placement of electrodes on model).
I changed the description and presentation of the table. I tried to include practical application in both the introduction and the discussion. I have also provided the referencing literature. I have highlighted the typical medical applications, and the fact that the diagnosis depends on the magnitude of the amplitude and/or latency. This is a very important aspect of the evaluation, diagnosis of almost any CNS disease.
I changed my sentence regarding "Accuracy" because my next paper will be about the application of neural networks and I will not combine that in this paper. I apologize for such a mistake.
Not sure if I understood correctly, but I included the computer system used and described what software was used (Windows, ANT). I included the practical application of using filters in the introduction as well as in the discussion. I added a description and literature.
Reviewer 3 Report
The manuscript presents some experimental results of EEG signals and their filtering. There is no novelty in it. You present so detailed data that it could be an experimental report instead of the paper. Additionally, the conclusions are extremely short and don't give any novelty.
I have noticed some problems with English and editorial remarks as well.
Author Response
Responses to Reviewer 3 Comments:
Thank you for all the comments.
I used the "Track Changes". Changes are visible throughout the document.
Changes: I edited the entire document for comments from all reviewers.
The paper was proofread by a certified linguist.
Changes:
It may seem to readers that this is just another paper that deals with artifacts removal, filtration, etc. There are many papers in this area that it is difficult to do something new. However, the Authors show the aforementioned problems from the point of view of amplitude/latency analysis.
The impact of filters/software/analysis is not often presented and specifically considers ERP and motor cortex area. I wanted more researchers to be able to see the results and refer to their analyses or future study. I personally miss this type of publication, hence the idea to present the analysis in this way.
Reviewer 4 Report
This manuscript attempted to figure out the impact of filters on EEG-ERP testing. However, the quality of the submitted manuscript is not reaching the level of publication.
Major points:
1) The authors mentioned that “Different methods are used to remove each type of artifact.” In Introduction without any details. Readers cannot find what kinds of filters are used for each purpose, and why the current state is not appropriate. Also, no related work reported.
2) Objective of this study is not clear.
3) Presentation, analysis, discussion of this manuscript is not ready for submission.
4) The number of subjects, kinds of experiments are very limited.
Minor points:
1) Style of the title does not look appropriate. Please capitalize the first character of each meaningful word in the title. Also, the period between the main title and sub-title (.) can be replaced with a colon(:).
2) What is the meaning of the first sentence of this manuscript??
3) There are several incomplete sentences. Some sentences use different fonts. A rigorous review of the manuscript is required before submission.
4) Section number (2.8) was wrongly described.
5) Style of text/writing is not consistent throughout the manuscript. Even the text from MDPI template is still included in this manuscript. Not ready for submission.
6) References for some articles are not correctly provided. For example, the reference for the dataset used in this study does not provide enough information to find it.
7) Presentation of Table 1 is not good. Not readable.
Author Response
Responses to Reviewer 4 Comments:
Thank you for all the comments.
I used the "Track Changes". Changes are visible throughout the document.
Changes: I edited the entire document for comments from all reviewers.
The paper was proofread by a certified linguist.
Changes:
1. Added information in both introduction and discussion. Added related work.
2. Aim of this study was Modified.
3. Revisions were made throughout the manuscript. The presentation and description of data/results were modified.
4. The experiment and the results of the experiment were performed and described in previous works. In this work, the authors wanted to show the "technical" value. We focused on the presentation of possible results, different approaches to the analysis, which has a great impact on the diagnosis.
-Style of the title changed according to comments.
-The first sentence has been changed.
-Text formatted according to guidelines.
- Chapter number corrected.
-Style corrected.
-I added DOI to the publication. All references were created using Zotero.
-Table was modified. A ready-made template was used (template for "Sensors").
Round 2
Reviewer 2 Report
In this new submission authors have included some practical applications about their approach. Honestly, I expected a point by point response letter to my comments. I could not find, in the document, the response to some of my questions. Could you provide the lines in which my comments were answered?
- Features of the PC used, RAM, micro, GPU, etc. mentioned in line 152
- Where is Figure 3?
- What happen with lines 154, 166-167 of previous versions?
Additional comments of current version are:
- I suggest moving lines 356-363 to conclusion section.
- Line 171 is incomplete
Author Response
Response to Reviewer 2
1. „Features of the PC used, RAM, micro, GPU, etc. mentioned in line 152”
Answer: I added the information in line 154.
2. Where is Figure 3?
Answer: I added Figure3.
3. What happen with lines 154, 166-167 of previous versions?
Answer: The text was modified according to comments from reviewers.
4. Additional comments of the current version are:
- I suggest moving lines 356-363 to the conclusion section.
Answer: Thank you very much for your attention. I changed.
- Line 171 is incomplete
Answer: I corrected.
5. Earlier Questions (1 review): ."..So, my point is, when these changes in amplitude or latency can be crucial for EEG-based systems performance?"
Response: If I understood the question correctly, I did not include the answer in the text. I agree with the reviewer's opinion that filters can introduce unwanted delay and amplitude attenuation of the original signal. However, if appropriate filter structures are used, then these parameters can be accurately determined and included in the final measurement results without significantly distorting the diagnostic information.
6. The text has been reviewed again for better readability.
7. I improved the results, they are now more readable and they present different approaches. I mainly wanted to show the differences (that may appear) between results.
Reviewer 3 Report
I don't have more comments.
Author Response
Response to Reviewer 3
The text has been reviewed again for better readability.
I have expanded the results and I think they are clearer now.
Reviewer 4 Report
The reviewer is not sure if Table 1, Figures 4/5, and the text regarding those parts do have consistent results. The authors mentioned that they used 8 electrodes, but Table 1 only presented part of them, without any reason provided. Also, Table 1 shows that there are filters like "raw", "bandpass", "notch", which is not consistent with the authors' statements that they used four different kinds of filters.
Figure 4/5 looks like they show only results on FIR filter, but in the text, the authors mentioned some results on the analysis between FIR and other filters which is not identifiable from the manuscript.
Please clarify these points to improve the readability of the paper.
Finally, the reviewer suggests the manuscript should be going through language editing for better readability.
Author Response
Response to Reviewer 4 Comments
Thank you for all your comments.
I've expanded the results and I think they are more readable now. You were right that Figures 4 and 5 were only for FIR. I left that graph and added others to make the paper more readable.
Table 1 was left for raw, bandpass, and notch, but in the rest of the paper, I showed another table (table 3) and graphs comparing the filters (to not put so many numerical values).
The text has been reviewed again for better readability